# Evidence of a one-dimensional thermodynamic phase diagram for simple glass-formers

H.W. Hansen[1], A. Sanz[1], K. Adrjanowicz[2], B. Frick [3] & K. Niss[1]

Glass formers show motional processes over an extremely broad range of timescales, covering more than ten orders of magnitude, meaning that a full understanding of the glass transition needs to comprise this tremendous range in timescales. Here we report simultaneous dielectric and neutron spectroscopy investigations of three glass-forming liquids, probing in a single experiment the full range of dynamics. For two van der Waals liquids, we locate in the pressure–temperature phase diagram lines of identical dynamics of the molecules on both second and picosecond timescales. This confirms predictions of the isomorph theory and effectively reduces the phase diagram from two to one dimension. The implication is that dynamics on widely different timescales are governed by the same underlying mechanisms.

[1] Glass and Time, IMFUFA, Department of Science and Environment, Roskilde University, Postbox 260DK-4000 Roskilde, Denmark. [2] Institute of Physics, University of Silesia, ul. Uniwersytecka 4, 40-007 Katowice, Poland. [3] Institut Laue-Langevin, 71 avenue des Martyrs, CS 20156, 38042 Grenoble Cedex 9, France. Correspondence and requests for materials should be addressed to K.N. (email: kniss@ruc.dk)

The glass transition plays a central role in nature as well as in industry, ranging from biological systems such as proteins and DNA to polymers and metals[1–3]. Yet, the fundamental understanding of the glass transition which is a prerequisite for optimized application of glass formers still lacks[4–7]. Glasses are formed when the molecular motions of a liquid become so slow that it effectively becomes a solid. When the glass transition of a liquid is approached, the dynamics of the molecules spreads out to several processes covering more than ten orders of magnitude. There are at least three overall contributions to the dynamics of glass-forming liquids, wich are as follows: (1) vibrations, (2) fast relaxations on picosecond timescales, and (3) the structural alpha relaxation, which has a strongly temperature-dependent timescale. The glass transition occurs when the alpha relaxation is on a timescale of hundreds of seconds and therefore completely separated from the two fast contributions to dynamics. Nonetheless, both theoretical[8,9] and experimental results[10–16] have suggested that fast and slow dynamics are intimately connected. A complete understanding of the glass transition therefore necessitates a full understanding of all these dynamic processes.

During the past 15 years, pressure has increasingly been introduced to study dynamics of glass-forming liquids in order to disentangle thermal and density contributions to the dynamics. The most striking finding from high-pressure studies is that the alpha relaxation, both its timescale and spectral shape is, for a large number of different liquids, uniquely given by the parameter $\Gamma = \rho^\gamma/T$, where $\gamma$ is a material specific constant[17–20]. This scaling behavior can be explained by isomorph theory[21,22], which, moreover, predicts the value of $\gamma$[23]. The fundamental claim of isomorph theory is the existence of isomorphs. Isomorphs are curves in the phase diagram along which all dynamic processes and structure are invariant. Put in other words, the phase diagram is predicted to be one-dimensional with respect to structure and dynamics on all timescales, with the governing single variable being $\Gamma$. Isomorph theory has been very successful in describing Lennard–Jones type computer-simulated liquids, e.g., refs. [24,25]. Experimental studies of isomorph theory predictions require high-precision high-pressure measurements and are still limited[23,26–28]. Consequently, it remains open to what extend isomorph theory holds for real molecular liquids.

The only systems that obey isomorph theory exactly are those with repulsive power law interaction potentials, which do not, of course, describe real systems. Hence, isomorph theory is approximate in its nature and expected to work for systems without directional bonding and competing interactions[22]. With this in mind, we have studied the dynamics on three well-studied glass formers representing non-associated liquids and liquids with directional bonding, two van der Waals bonded liquids (vdW-liquids): PPE (5-polyphenyl ether) and cumene (isopropyl benzene), and a hydrogen bonding (H-bonding) liquid: DPG (dipropylene glycol).

In this work, we use a high-pressure cell for simultaneous measurements of the fast dynamics by neutron spectroscopy and the alpha relaxation by dielectric spectroscopy to demonstrate that for the studied vdW-liquids, the three mentioned distinct dynamic components are invariant along the same lines in the phase diagram. This is direct experimental evidence for the existence of isomorphs, and it shows that the phase diagram of these simple vdW-liquids is one-dimensional with respect to dynamics. Unlike the scaling behavior of the alpha relaxation dynamics, which is often found to hold surprisingly well in H-bonding systems[27–29], we only find invariance of the fast relaxational and vibrational dynamics on picosecond timescales in the vdW-liquids. For the single investigated H-bonding system, DPG, we make a different observation, as expected based on isomorph theory.

## Results

**Simulatenous dielectric and neutron spectroscopy.** Dynamics from picosecond to kilosecond cannot be measured with one single technique; several complementary techniques are required. A glass-forming liquid is in metastable equilibrium and the dynamics is very sensitive to even small differences in pressure and temperature. The high viscosity of the liquid close to the glass transition temperature, $T_g$, makes the transmission of isotropic pressure non-trivial, as pressure gradients are easily generated. In order to ensure that the different dynamics are measured under identical conditions, we have therefore developed a cell for doing simultaneous dielectric spectroscopy (DS) and neutron spectroscopy (NS) under high pressure (Fig. 1)[30]. The experiments were carried out on spectrometers at the Institut Laue-Langevin (ILL) on the time-of-flight (TOF) instruments IN5 and IN6. The different NS instruments access different timescales with IN5 and IN6 giving information on the ~10 ps scale, while a backscattering (BS) instrument like IN16 accesses ~1 ns dynamics. DS provides fast (minutes) and high accuracy measurements of the dynamics from microsecond to 100 s.

The dynamics measured with the different techniques are illustrated in Fig. 1a, b for PPE. The center panels (a) and (b) are sketches of the incoherent intermediate scattering function, $I(Q, t)$, while the top and bottom panel show raw data. At $T_g$ (Fig. 1a), no broadening is observed on nanosecond timescales (IN16) corresponding to a plateau in $I(Q, t)$, on picosecond timescales from IN5, we observe contributions from fast relaxational processes and vibrations, whereas the alpha relaxation is seen in DS at much longer timescales, a difference of more than 10 orders of magnitude. As the temperature is increased, the processes merge (Fig. 1b), and relaxation dominates the signal in all three spectrometers.

**Picosecond dynamics at the glass transition.** The focus in this paper is on the picosecond dynamics measured on IN5 and IN6, while the IN16 data are only used as an illustration in Fig. 1. The IN5 (cumene and PPE) and IN6 (DPG) data from a set of temperature–pressure state points are presented in Fig. 2. We observe the same trend for all spectra for all values of wave vector $Q$ (Supplementary Fig. 2), and have summed over $Q$ to improve statistics. All spectra are shown on the same $S(\tilde{\omega})$-axis. Motivated by isomorph theory, the energy scale is shown in reduced units, effectively $\tilde{\omega} = \omega \rho^{-1/3} T^{-1/2}$ [22]. The effect of scaling is small, though visible, in the studied range of $\rho$ and $T$. The data is shown on an absolute energy scale in the Supplementary Fig. 1 and more details on the scaling is given in the Methods section.

For all three samples in row (a) Fig. 2, which shows dynamics in the liquid, we observe the two extreme scenarios sketched in Fig. 1. At low pressure, relaxational contributions are dominating (Fig. 1b). At the glass transition, we find only fast relaxational and vibrational contributions (Fig. 1a). The fast relaxational contributions decrease in the glassy state (Fig. 2b), leaving the excess vibrational density of states, which shows up as the so-called Boson peak[31], as the dominant contribution (black full lines in Fig. 2b). For all three samples, we observe different dependencies on temperature and pressure for the three contributions to the dynamics, such that their relative contributions vary along both isobars and isotherms.

All glass formers have isochrones, which are lines in the $(T, P)$-phase diagram with constant alpha relaxation time, $\tau_\alpha$. If isomorphs exist in a liquid, these coincide with the isochrones, since all dynamic processes on all timescales and of all dynamic variables are invariant along an isomorph. Thus, experimentally, we can identify candidates for isomorphs by the isochrones. We use DS effectively as a 'clock', which identifies the alpha relaxation

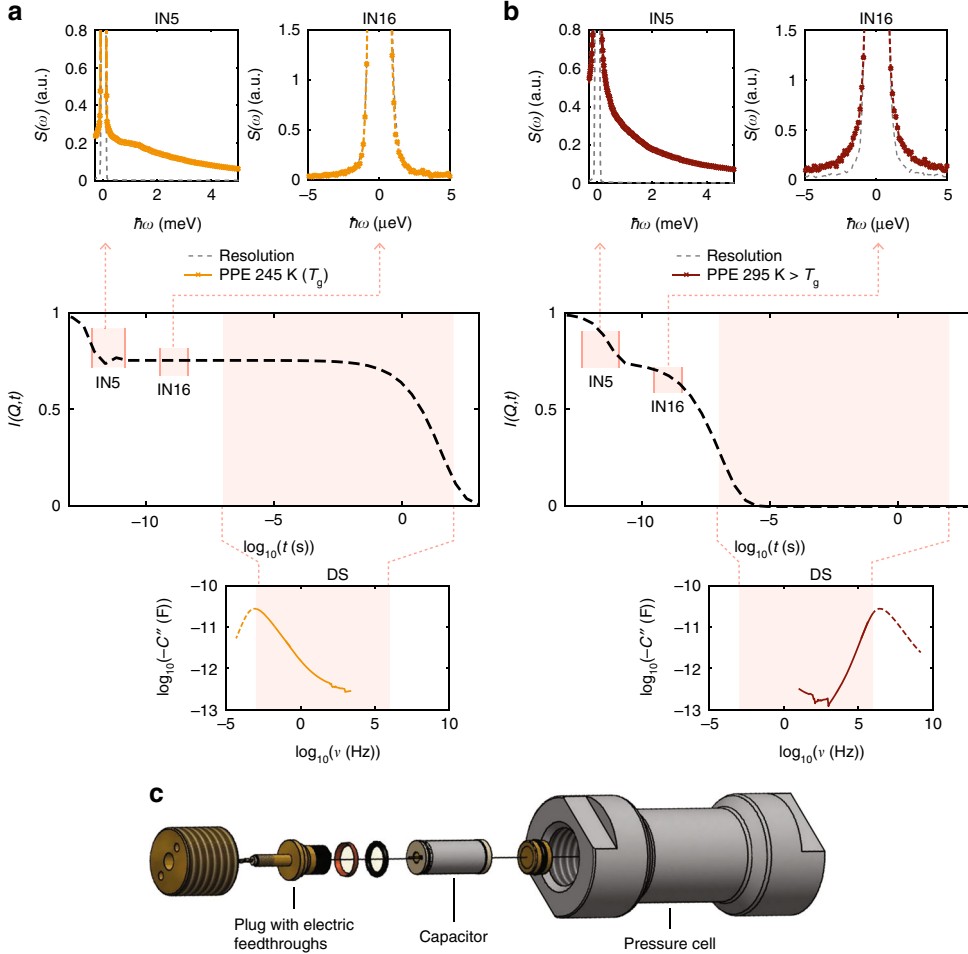

**Fig. 1** Dynamics probed with dielectric (DS) and neutron spectroscopy. **a**, **b** Sketch the incoherent intermediate scattering function $I(Q, t)$ in the center panel. Top panels show raw spectra measured on the time-of-flight and backscattering spectrometers IN5 and IN16, respectively, and bottom panels show spectra from DS, all measured on PPE. Dashed lines in DS are from time-temperature superpositon (TTS). **a** Dynamics at the glass transition $T_g$ where there is separation of timescales. At picosecond timescales we observe fast relaxation and vibrations. At nanosecond timescales there is no relaxation and the alpha relaxation is only visible in DS. **b** Dynamics in the liquid well above $T_g$. Relaxation dominates the signal in all three spectrometers. **c** Drawing of components of simultaneous high-pressure dielectric and neutron spectroscopy cell

time from the dipole–dipole correlation function, while simultaneously measuring the picosecond dynamics with incoherent neutron scattering probing the particle self-correlation.

Row (c) in Fig. 2 shows the picosecond dynamics measured along the glass transition isochrone $T_g(P)$ defined as when $\tau_\alpha = 100$ s found from DS. We observe superposition of the spectra at the picosecond timescale, thus invariance of the dynamics, for the two vdW-liquids (PPE and cumene). This is in agreement with the prediction of isomorph theory and it is particularly striking because at $T_g(P)$, fast relaxational and vibrational motion are completely separated in timescale from the alpha relaxation as illustrated in Fig. 1a. In contrast, for the H-bonding liquid (DPG), we find a clear shift towards higher energy and an intensity decrease of the Boson peak along the $T_g(P)$ isochrone. The lack of superposition in the H-bonding system demonstrates that the superposition seen in the vdW-liquids is non-trivial. Thus, the superposition observed in the vdW-liquids is a genuine signature of the isomorphs in these liquids.

**One-parameter phase diagram**. To compare more state points, including shorter alpha-relaxation time isochrones ($\tau_\alpha < 1\,\mu$s), isotherms and isobars, we plot the corresponding inelastic intensities at a fixed reduced energy ($\tilde{\omega} = 0.06$) as a function of

temperature and pressure (Fig. 3a, b). Along the investigated isochrones, the inelastic intensity at $\tilde{\omega} = 0.06$ or $t \sim 1$ ps is found to be invariant for the vdW-liquids. Again, the H-bonding liquid behaves differently, and its picosecond dynamics varies along the isochrones. This confirms the isomorph prediction of spectral superposition of fast relaxation, vibrations/Boson peak and alpha relaxation for the vdW-liquids: both when there is timescale separation and when the processes are merged.

The power of isomorph theory is that the phase diagram becomes one-dimensional; the dynamics only depend on which isomorph a state point is on and not explicitly on temperature and pressure (or density). Since isomorphs are isochrones, the other dynamic components should become a unique function of the alpha relaxation time. In Fig. 3c we plot the intensity of the picosecond dynamics along isotherms and isobars as a function of the dielectric alpha relaxation time at fixed reduced energies ($\tilde{\omega} = 0.02, 0.04, 0.06, 0.08, 0.1$). The data from each energy collapses in this plot, illustrating that, as predicted, all the dynamics is governed by one parameter.

## Discussion

Previously suggested connections between fast and slow dynamics, e.g., via the temperature dependence of properties[7,14], often

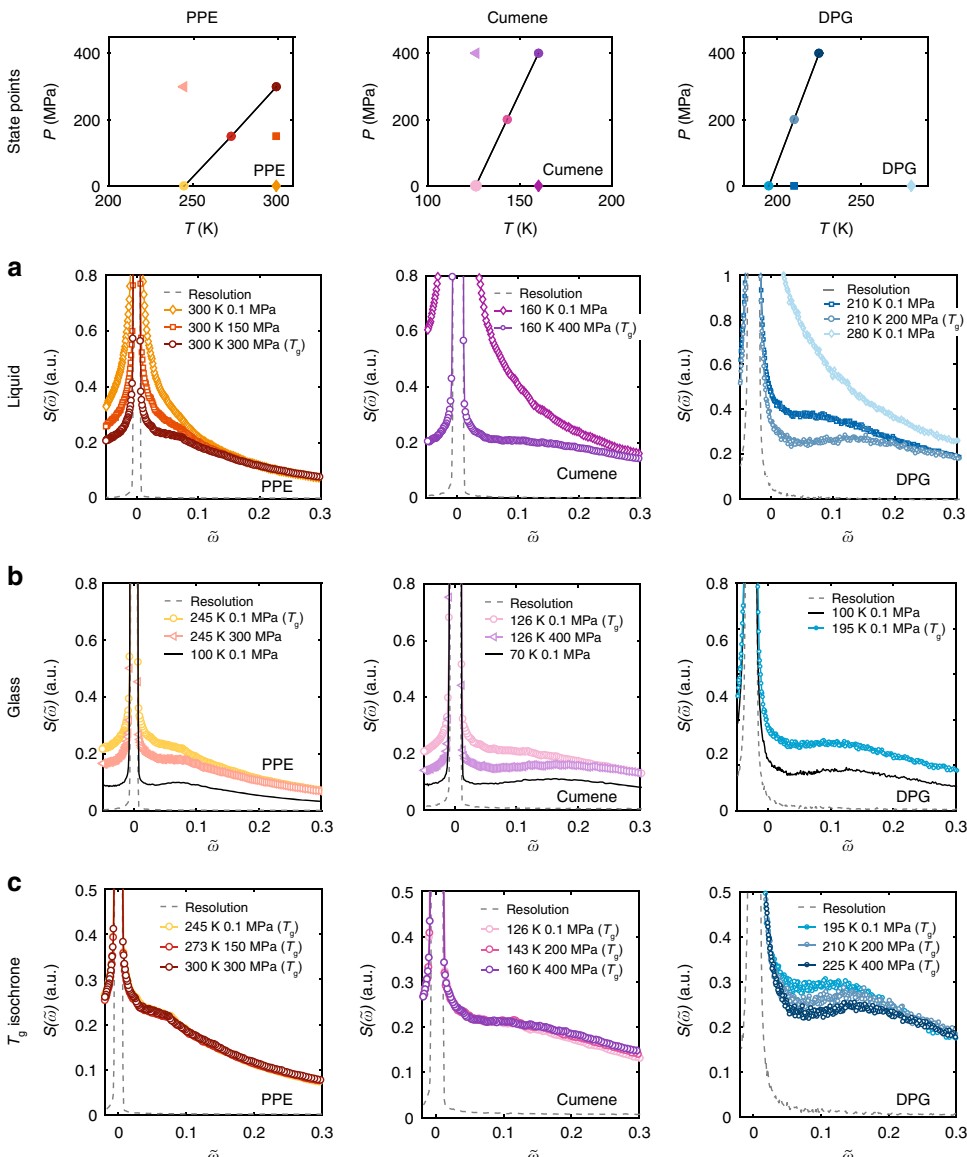

**Fig. 2** Picosecond dynamics at various state points for PPE, cumene and DPG. Top panel: $(T, P)$-phase diagram showing the state points of the spectra in row **a**–**c**. The black lines correspond to the glass transition isochrones with $\tau_\alpha = 100$ s. All spectra are plotted in reduced units, $\tilde{\omega} = \omega \rho^{-1/3} T^{-1/2}$ and summed over $Q$. **a** Spectra in the liquid: at low pressure, we primarily observe relaxation. As pressure is increased, reaching the glass transition, only fast relaxation and vibrations are left. **b** Spectra in the glass: leaving the glass transition, going deeper into the glass, fast relaxations disappear and only vibrations are left. **c** Along the glass transition isochrones, there is superposition of dynamics for the two van der Waals liquids, PPE and cumene, while in the hydrogen bonding liquid, DPG, there is a clear shift in the data with pressure. PPE and cumene were measured on IN5 and DPG on IN6. No scaling has been done on the $y$-axis, i.e. all spectra are shown on the same $S(\tilde{\omega})$ scale

suggest a causality, where the fast dynamics controls the slow dynamics. In contrast, the connection between slow and fast dynamics shown in this work does not tell us if one controls the other or if they are simply controlled by the same underlying mechanism.

Isomorph theory is approximate in its nature, and the isomorphs of real physical systems are approximate. The fact that the isomorph prediction works for dynamics which is separated in timescale by more than 10 orders of magnitude tells us that whatever governs this dynamics is controlled by properties of the liquid that obey the isomorph scale invariance.

Most of the computer simulation tests on isomorph theory have been performed on atomic systems or systems with rigid bonds. When attempts are done to test the theory in molecular models with springs between the atoms, i.e., allowing for

intramolecular vibrations, the basic assumption of the isomorph theory no longer applies, and isomorphs are not predicted to exist[32]. However, in a recent paper[33], a method was developed to separate intramolecular modes from the vibrations in a computer-simulated molecular liquid. By only considering the vibrations associated with intermolecular interactions, it is possible to recover the isomorphs and show that vibrations and alpha relaxation are invariant along the same lines in the phase diagram. The interpretation of this result is that: (a) fast and slow dynamics are governed by the same potential energy landscape, and (b) it is, in fact, this energy landscape that is invariant (in reduced units) along the isomorphs.

In line with this, our interpretation is that the three contributions to dynamics we study, fast relaxation, Boson peak and alpha relaxation, are all controlled by the same energy landscape,

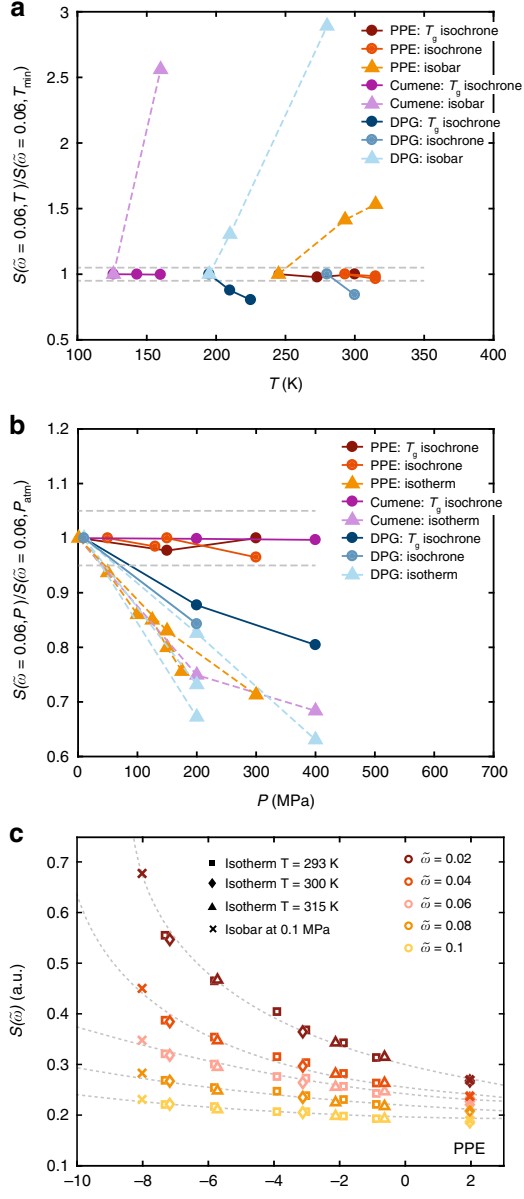

**Fig. 3** Comparison of state points via the inelastic intensity at fixed reduced energies. **a**, **b** Inelastic intensity at fixed reduced energy ($\tilde{\omega} = 0.06$) along isochrones, isotherms and isobars from the spectra in Fig. 2 plotted **a** as a function of temperature normalized to value at lowest temperature, **b** as a function of pressure normalized to the value at ambient pressure. **c** Inelastic intensity for fixed reduced energies for PPE as a function of $\tau_\alpha(T, P)$ for two isotherms, 293 K (squares), 300 K (diamonds), 315 K (triangles) and an isobar (crosses) for the reduced energies $\tilde{\omega} = 0.02, 0.04, 0.06, 0.08, 0.1$ covering more than ten orders of magnitude in relaxation time. All lines are guides for the eye

and that the way this energy landscape is visited is governed by a single parameter $\Gamma = \rho^\gamma/T$. For this to be consistent with the computer simulations results mentioned above, it means that all the dynamics we study is governed by intermolecular, and not intramolecular interactions, which is also what we expect based on early experimental studies, where intra- and intermolecular modes are separated[34].

It is clear from the spectra in Fig. 2 that fast relaxation and Boson peak are different in nature because their relative

intensities vary along isotherms and isobars, and that these two types of fast dynamics are distinctively different from the structural alpha relaxation. Yet, all of these dynamic features are controlled by the single parameter $\Gamma = \rho^\gamma/T$. There is a consensus that the alpha relaxation is cooperative, fast relaxations are normally understood as cage rattling, whereas it has been heavily debated whether or not the Boson peak is localized[31,35–37].

Our finding implies that a universal theory for the glass-transition, including the description of fast dynamics, needs to be consistent with the one-dimensional phase diagram that comes out of isomorph theory. This statement is what is sometimes referred to as the isomorph filter—it means that a property of the liquid, which is claimed to control dynamics has to be invariant along isomorphs.

## Methods

**Materials.** Isopropyl benzene (cumene) and dipropylene glycol (DPG) were purchased from Sigma Aldrich, and 5-polyphenyl ether (PPE) was purchased from Santolubes. All three samples were used as acquired.

The glass transition for the three samples is found from dielectric spectroscopy and defined as when the maximum of the loss peak corresponds to $\tau_\alpha = 100$ s, where $\tau_\alpha = 1/(2\pi\nu''_{max})$. At atmospheric pressure, the glass transition temperature $T_g$ is 245 K for PPE, 126 K for cumene, and 195 K for DPG.

The traditional way of quantifying how much the alpha relaxation time, or the viscosity, as a function of temperature deviates from Arrhenius behavior is given by the fragility[38], defined as:

$$m = \frac{d\log_{10}\tau_\alpha}{d(T_g/T)}\bigg|_{T_g}. \tag{1}$$

For the two van der Waals liquids, the fragility at ambient pressure is $m \approx 80$ for PPE[39] and $m \approx 70$ for cumene[15,40]. The hydrogen-bonding liquid DPG has fragility $m \approx 65$[41,42].

**Experimental details.** Dynamics studied with neutron spectroscopy is well suited for a test of isomorph theory as there are known to be large changes in the dynamics on sub-nanosecond timescales for the same range of density and temperature changes as those that change the alpha relaxation time with some orders of magnitude. The static structure factor on the other hand changes very little, especially when plotted in reduced units.

All experiments were carried out at the Institut Laue-Langevin (ILL) on the time-of-flight instruments IN5 and IN6. In neutron spectroscopy, the different instrumental energy resolutions give access to different dynamical timescales; the coarser the energy resolution, the faster the time window accessible: $\Delta E_{res} \approx 0.1$ meV on IN5 and IN6 corresponds to ~10 ps. Dielectric spectroscopy (DS) provides fast (minutes) and high accuracy measurements of the dynamics from microsecond to 100 s. The same protocol has been used for all three samples, where pressure has been applied in the non-viscous liquid and then cooled to the state point of interest. The pressure transmission was carefully monitored with DS. The precision in pressure is determined by the pressure tolerance of the compressor, which is the setpoint ±3 MPa. Stability in temperature can be held at a fixed temperature ±0.1 K. The absolute uncertainty is a few degrees, but we use DS to identify isochronal state points. Details on the high-pressure cell for doing simultaneous dielectric and neutron spectroscopy can be found in the forthcoming publication[30].

In Fig. 2, we presented the data on PPE and cumene from IN5 and on DPG from IN6 (hence the difference in statistics). All the data were measured with a wavelength of 5 Å and an energy resolution ~0.1 meV, corresponding to a timescale of ~10 ps. All spectra have been corrected in the conventional way by normalizing to monitor and vanadium, subtracting background, and correcting for self-shielding, self-absorption and detector efficiency using LAMP, a data treatment program developed at the ILL. The data have then been grouped for constant wave vector $Q$ up to 8 meV in steps of 0.1 Å⁻¹ in the range 1.2–1.9 Å⁻¹ for the IN5 data (PPE and cumene) and in the range 1.2–1.7 Å⁻¹ for the IN6 data (DPG) and are presented in Fig. 2 as a sum over $Q$. The data are shown as measured in Supplementary Fig. 1 as a function of energy transfer on an absolute energy scale. The energy transfer range presented is up to $\hbar\omega = 5$ meV, as there is only little change in the shape at higher energy transfers. No scaling has been done on the $y$–axis in $S(\omega)$ of any of the spectra, i.e. all spectra are plotted on the same scale. Comparing Supplementary Figs. 1 and 2, the effect of plotting data in reduced units is visible at higher energy transfer. The same picture is observed for all spectra for each value of $Q$, and we have therefore summed over $Q$ in the data shown in Fig. 2 to improve statistics. An example of spectra along the glass transition at different $Q$ for PPE and DPG is shown in Supplementary Fig. 2.

**Reduced units**. According to isomorph theory, the relevant scale to look at is in reduced units[22]. The reduced energy units used in Fig. 2 and Supplementary Fig. 2 are given by

$$\tilde{\omega} = \omega t_0 = \omega \rho^{-1/3} \sqrt{m/(k_B T)}, \tag{2}$$

where $\omega$ is the energy transfer which is equivalent to the frequency except a factor of $\hbar$. $k_B$ is Boltzmann's constant and $T$ is temperature. Here, $\rho$ is the number density and $m$ is the average particle mass, the latter assumed constant. We set all the constants to one since these do not affect the scaling, $\hbar = m = k_B = 1$. Effectively, the scaling becomes

$$\tilde{\omega} = \omega \rho^{-1/3} T^{-1/2}, \tag{3}$$

where $\rho$ is now the volumetric mass density.

Just like reduced energy units, wave vector or momentum transfer $Q$ should also be presented in reduced units:

$$\tilde{Q} = Q \rho^{-1/3} \tag{4}$$

But as the density changes are in the percent range in this study, scaling of $Q$ will be around 1% and will be within the uncertainty of the data and is therefore neglected.

When we find the glass transition isochrone, this should also be done from a reduced unit isochrone. However, the difference between the scaling factors from atmospheric pressure to the high pressure state point is for all three samples 1.2 or less, this corresponds to a relative shift of the dielectric curves of less than 0.1 decade. In practice, this is less than our precision in pressure allows us to fix.

Plotting Fig. 3c as a function of the alpha relaxation time on a reduced timescale instead of on an absolute timescale does not change the visual representation of the figure. Figure 3c is therefore plotted on an absolute timescale, because this conveys more information about the state of the system.

**Calculating density**. Equations of state (EOS) have been used to calculate the temperature and pressure dependence of the density. For cumene, the EOS is from[40]. For DPG, the EOS is taken from ref. [43]. For PPE, a fit to the Tait equation from PVT data from ref. [44] has been used to obtain the density:

$$\rho(T,P) = \left( V_0 \exp(\alpha_0 T) \left\{ 1 - C \ln \left[ 1 + \frac{P}{b_0 \exp(-b_1 T)} \right] \right\} \right)^{-1}, \tag{5}$$

where $\rho$ is in g/cm$^3$ and equal to $1/V_{sp}$, the specific volume, $P$ is pressure in MPa and $T$ is temperature in °C. The fitting parameters are $V_0 = 0.82$, $\alpha_0 = 6.5 \cdot 10^{-4}$, $C = 9.4 \cdot 10^{-2}$, $b_0 = 286$ and $b_1 = 4.4 \cdot 10^{-3}$.

For all three samples, the density changes are in the percent range in the temperature–pressure range of this study. The PVT data and EOS are only used for scaling on the energy axis of the data, where the reduced energy units contain the cubic root of the density. Hence, the scaling is in practice mainly with temperature, and the use of EOS therefore does not alter the overall conclusion.

**Data availability**. The data presented in this paper can be obtained from the "Glass and Time" data repository at glass.ruc.dk/data.

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

## Acknowledgements

This work was supported by the Danish Council for Independent Research (Sapere Aude: Starting Grant). We gratefully acknowledge J. Dyre and T. Schrøder for fruitful discussions and the workshop at IMFUFA and the SANE group at the ILL for technical support.

## Author contributions

The project was conceived by K.N. The experiments were managed by H.W.H. and carried out by H.W.H., A.S., K.A., B.F. and K.N. The data analysis was performed by H.W.H. The manuscript was written by H.W.H. and K.N. with input from A.S., K.A. and B.F.

## Additional information

**Competing interests:** The authors declare no competing financial interests.

