## [Peer Review File · Nature Communications]

Reviewers' comments:

Reviewer #1 (Remarks to the Author):

The authors are members of a large group which achieved the construction of an ingenious neutron scattering pressure cell capable of simultaneous dielectric measurements. With this sample holder, they measured three molecular liquids at their glass transition, two van der Waals bonded ones and a hydrogen bonded one. The fast relaxations and the boson peak scale according to the predictions of the isomorph theory in the two van der Waals substances, but fail to do so in the hydrogen bonded liquid.

This is an experimentum crucis. There is no doubt at all that it merits publication in Nature. But though the work is already well presented, with appropriate supplementary material on the pressure cell and other experimental details, I think that some polishing could still augment the brilliance of this diamond:

1. The reader gets stuck at the introduction of the scaled omega units (first paragraph page 5). Did you not achieve the same τ_α experimentally? And now you suddenly change the time scale?

Here you need to give numbers for the scaling factors. And you need to explain that these scaling factors are practically irrelevant for the determination of the glass temperature.

2. In the middle of page 6, you conclude that "the superposition observed in the vdW-liquids is a genuine signature of the isomorphs in these liquids." Right, this is the main conclusion of the paper.

But here is an opportunity to discuss the physics of the failure of the isomorph theory in dipropylene glycol, a molecule with two hydrogen bonds, which fix two of its three degrees of freedom. This leaves room for one van-der-Waals vibration per molecule.

It seems that these van-der-Waals vibrations tend to concentrate in the boson peak, similar to the case of selenium (Phillips et al, PRL 63, 2381 (1989)) making it more pressure-dependent than τ_α , which in its turn requires the breaking of hydrogen bonds. If I am right, the Grüneisen parameter of the boson peak should be approximately the same as the one in the two other substances. The authors could check that from the existing data.

3. From there, it would be appropriate to point out that the application of the isomorph theory to glass transition experiments requires the same Grüneisen parameters for all relevant degrees of freedom. The example of the two van der Waals substances shows that the rather harmonic high frequency molecular vibrations are irrelevant.

To summarize, I believe that the physical significance of this exceptionally beautiful result merits a more detailed discussion.

Reviewer #2 (Remarks to the Author):

In the last 10 years, Dyre and coworkers have developed the isomorph theory and tested it extensively with computer simulations. When a system obeys the isomorph theory, the two dimensional space of P and T is functionally reduced to a one dimensional space. This is an enormous simplification. This is a useful idea that is having a considerable impact on the field. I anticipate that the impact can be much more important as careful experimental work is used to test the isomorph idea.

The current work is a major advance in the testing of isomorph theory for experimental systems. The authors identify potential isomorphs by finding two values of (P,T) that have the same average relaxation time. If the system follows the theory, then all dynamical properties at these two state point should be identical. The current data is consistent with this prediction for two of the three systems studied, with the third system clearly not following the prediction. The use of two techniques that measure dynamics on two very different times scales (seconds and picoseconds!) is a key feature of the present work – this has not been done previously and this

makes the test much more rigorous. There have been previous indications of connections between short time and long time dynamics, but the present demonstration is a substantial step forward. This is a very important paper and I support its publication in Nature Communications after the authors have had an opportunity to consider changes in light of my comments below.

1) The introduction makes clear when the isomorph theory is applicable that there is also a relationship between the structures at the two state points. This is not tested in the current work, as far as I can see. I am surprised that they did not also use neutron scattering to look at structure. The authors should comment on this.

2) I am struggling a bit with the idea that the vibrational dynamics will also have a simple relationship at the two isomorph state points. I accept the authors view that they have shown that this works for some vibrational properties in this manuscript. Molecular systems have internal vibrational modes – are the authors claiming that these high frequency modes also obey the isomorph relationship? If not, could the authors clarify what types of vibrational processes are relevant for isomorph theory and how they know that they are looking at this type of vibrational mode? I am concerned that someone coming to the paper from outside the glass community will see molecules and think that “vibrations” mean something different than the authors intend.

3) There are various mentions of IN6 and IN 16. I wonder if one of these is a typo? I think only one is described in the experimental details.

Reviewer #3 (Remarks to the Author):

I enjoyed reading the manuscript and liked the way the Authors wrote it. For the first time simultaneous neutron and dielectric spectroscopy investigations of three different (both van der Waals and H-bonded) glass-forming liquids in T-P thermodynamic space were performed. It means that in single experiment fast and slow dynamics was simultaneously probed. Based on these measurements, the Authors proved experimentally that isomorph prediction also works for fast dynamics. It is very fundamental finding!

In my opinion this manuscript should be definitively published in Nature Communications.

Reply to reviews on manuscript “Evidence of a one-dimensional thermodynamic phase diagram for simple glass-formers”

First of all we would like to thank all referees for their positive reviews and for very useful comments and suggestions. Below are our replies to the specific comments from reviewer 1 and 2 (reviewer 3 had no comments) including details on how we have changed the manuscript.

Replies to reviewer 1,

1. The reader gets stuck at the introduction of the scaled omega units (first paragraph page 5). Did you not achieve the same tau_alpha experimentally? And now you suddenly change the time scale? Here you need to give numbers for the scaling factors. And you need to explain that these scaling factors are practically irrelevant for the determination of the glass temperature.

Changes made The section about reduced units has been moved from supplementary material into the Methods section, which is part of the main paper in the format of Nature Communications. The first time we mention reduced units we refer to this section. Moreover, the section on reduced units has been extended to include a paragraph on how it affects the alpha relaxation time as determined from the dielectric spectra. In total the section now reads:

According to isomorph theory, the relevant scale to look at is in reduced units¹¹. The reduced energy units used in Fig. 2 and Supplementary Information Fig. S2 are given by

$$\tilde{\omega} = \omega t_0 = \omega \rho^{-1/3} \sqrt{m/(k_B T)} \quad (1)$$

where ω is the energy transfer which is equivalent to the frequency except a factor of \hbar . k_B is Boltzmann’s constant and T is temperature. Here, ρ is the number density and m is the average particle mass, the latter assumed constant. We set all the constants to one since these do not affect the scaling, $\hbar = m = k_B = 1$. Effectively, this becomes

$$\tilde{\omega} = \omega \rho^{-1/3} T^{-1/2}, \quad (2)$$

where ρ is now the volumetric mass density.

Just like reduced energy units, wave vector or momentum transfer Q should also be presented in reduced units:

$$\tilde{Q} = Q \rho^{-1/3} \quad (3)$$

But as the density changes are in the percent range in this study, scaling of Q will be around 1% and will be within the uncertainty of the data and is therefore neglected.

When we find the glass transition isochrone, this should also be done from a reduced unit isochrone. However, the difference between the scaling factors from atmospheric pressure to the high pressure state point is for all three samples 1.2 or less, this corresponds to a relative shift of the dielectric curves of less than 0.1 decade. In practice, this is less than our precision in pressure allows us to fix.

Plotting Fig. 3c as a function of the alpha relaxation time on a reduced timescale instead of on an absolute timescale does not change the visual look of the figure. Fig. 3c is therefore plotted on an absolute timescale, because this conveys more information about the state of the system.

2. In the middle of page 6, you conclude that " the superposition observed in the vdW-liquids is a genuine signature of the isomorphs in these liquids." Right, this is the main conclusion of the paper. But here is an opportunity to discuss the physics of the failure of the isomorph theory in dipropylene glycol, a molecule with two hydrogen bonds, which fix two of its three degrees of freedom. This leaves room for one van-der-Waals vibration per molecule. It seems that these van-der-Waals vibrations tend to concentrate in the boson peak, similar to the case of selenium (Phillips et al, PRL 63, 2381 (1989)) making it more pressure-dependent than τ_{α} , which in its turn requires the breaking of hydrogen bonds. If I am right, the Gruneisen parameter of the boson peak should be approximately the same as the one in the two other substances. The authors could check that from the existing data.

This comment has inspired some interesting discussions in our group and we do agree that there is more physics to be extracted. We look forward to continuing the discussion. One of the aims of the further work in the theoretical part of our group is in fact to work systematically with coarse-graining in order to establish pseudo-isomorphs, which are lines where the dynamics of certain degrees of freedom are invariant while other degrees of freedom are not invariant. This work has been started in a recent paper [J. Chem. Phys., 145, 241103 (2016)], where it is shown how intramolecular modes can be handled in computer simulations of simple molecular systems.

Regarding the specific idea of looking at the Gruneisen parameter of the boson peak, it is not simple to access. The Boson peak is not well resolved in the liquid, it can barely be separated from relaxations at T_g . Therefore one would have to look at the Gruneisen parameter in the glass - however in the glass we do not have PVT-information and we do not know what the density changes of the sample is. Moreover, while the Gruneisen parameter of one system should be the same for different modes, we do expect it to differ from system to system.

Changes made We have expanded and restructured the discussion inspired by the comments of the reviewers.

3. From there, it would be appropriate to point out that the application of the isomorph theory to glass transition experiments requires the same Gruneisen parameters for all relevant degrees of freedom. The example of the two van der Waals substances shows that the rather harmonic high frequency molecular vibrations are irrelevant.

This comment also instigated a series of discussions in our group in Roskilde. It touches upon some of the very fundamental aspects of isomorph theory. Yet, we believe that entering a discussion of the Gruneisen parameter will be too specialized for this paper. Especially because this point has not really been discussed in any previous theory papers that we can refer to.

Regarding the high frequency molecular vibrations, we agree that the scaling we find, at least the interpretation in terms of isomorph theory, implies that the boson peak and the fast relaxation we present are unaffected by intra-molecular modes. In fact, we believe that the fast dynamics of the van der Waals bonded liquids is governed by the shape of the potential energy surface which also governs alpha relaxation. This again relates to the recent JCP paper mentioned above [J. Chem. Phys., 145, 241103 (2016)] and we have added some discussion of this in the paper.

Changes made We have expanded and restructured the discussion inspired by the comments of the reviewers.

Replies to reviewer 2

1. The introduction makes clear when the isomorph theory is applicable that there is also a relationship between the structures at the two state points. This is not tested in the current work, as far as I can see. I am surprised that they did not also use neutron scattering to look at structure. The authors should comment on this.

This is a very good point. The challenge is that the structure factor changes little with the pressure and temperature changes that correspond to changes in the alpha-relaxation with orders of magnitude from one end of the dielectric range to the other. This is particularly true when recalling that isomorph predictions concern the behavior measured in reduced units, meaning that the trivial change in structure factor due to density has to be scaled out before comparing state points. The dynamics measured in $S(q, \omega)$, on the other hand, changes very clearly in the same P, T -range. Showing that structure is invariant along the isomorph is only interesting if it is NOT invariant in the rest of the phase diagram. This requires very high precision on the $S(q)$ measurements. Nevertheless this is something we hope to pursue in the future. A technicality is that the cell will also need to be modified - as one would ideally have more sample in the beam for $S(q)$ measurements. An alternative route that we are also considering is moving to X-rays. However, working in the relevant temperature with heavy cooling equipment and with pressure cells is easier with neutrons. In either case, the structure studies are beyond the scope of this work.

Changes made We have added a comment regarding this point in the method section.

Dynamics studied with neutron spectroscopy is well suited for a test of isomorph theory as there are known to be large changes in the dynamics on sub-nanosecond timescales for the same range of density and temperature changes as those that change the alpha relaxation time with some orders of magnitude. The static structure factor on the other hand changes very little, especially when plotted in reduced units.

2. I am struggling a bit with the idea that the vibrational dynamics will also have a simple relationship at the two isomorph state points. I accept the authors view that they have shown that this works for some vibrational properties in this manuscript. Molecular systems have internal vibrational modes – are the authors claiming that these high frequency modes also obey the isomorph relationship? If not, could the authors clarify what types of vibrational processes are relevant for isomorph theory and how they know that they are looking at this type of vibrational mode? I am concerned that someone coming to the paper from outside the glass community will see molecules and think that “vibrations” mean something different than the authors intend.

This is an important point and it relates to the comment made by reviewer 1, that more information could be extracted. It is intermolecular vibrations we have in mind, phonons, local modes, or what is some times referred to as cage rattling.

Changes made We have expanded and restructured the discussion inspired by the reviewers comments.

3. There are various mentions of IN6 and IN 16. I wonder if one of these is a typo? I think only one is described in the experimental details.

We thank the reviewer for pointing out that this was not clear. In fact data from IN16 are only used in Fig. 1. The point of including data from IN16 is to show that there is no signal at T_g in time range covered by IN16, which means that the alpha-relaxation is completely separated from the dynamics we measure at IN5 and IN6. IN5 and IN6 access dynamics on the same time scale. However, IN5 has higher flux, using one or the other is a question of where we were able to get beam time.

Changes made We have made the text more explicit regarding this point. It now reads:

The experiments were carried out on spectrometers at the Institut Laue-Langevin (ILL) on the time-of-flight (TOF) instruments IN5 and IN6. The different NS instruments access different timescales with IN5 and IN6 giving information on the ~ 10 ps scale, while a backscattering (BS) instrument like IN16 accesses ~ 1 ns dynamics. DS provides fast (minutes) and high accuracy measurements of the dynamics from microsecond to 100 s.

The dynamics measured with the different techniques are illustrated in Fig. 1a and b for PPE. The center panels of (a) and (b) are sketches of the incoherent intermediate scattering function, $I(Q, t)$, while the top and bottom panel show raw data. At T_g (Fig. 1a), no broadening is observed on nanosecond timescales (IN16) corresponding to a plateau in $I(Q, t)$, on picosecond timescales from IN5 we observe contributions from fast relaxational processes and vibrations, whereas the alpha relaxation is seen in DS at much longer timescales, a difference of more than 10 orders of magnitude. As the temperature is increased, the processes merge (Fig. 1b), and relaxation dominates the signal in all three spectrometers.

The focus in this paper is on the picosecond dynamics measured on IN5 and IN6, while IN16 data are only used as an illustration in Fig. 1.

REVIEWERS' COMMENTS:

Reviewer #1 (Remarks to the Author):

The authors have changed their paper to address the comments of the referees.

The changes regarding points 1) and 3) of Reviewer #2 and my own point number 1 seem completely satisfactory.

With regard to point 2) of Reviewer # 2, which is intimately related to my points 2 and 3, the authors admitted its relevance, but decided against its detailed discussion.

However, since all three referees agree that this is a fundamental result, I recommend publishing the paper in its present version.

Reply to reviews on manuscript “Evidence of a one-dimensional thermodynamic phase diagram for simple glass-formers”

For the second version of the paper we received one report, with the following comment.

The authors have changed their paper to address the comments of the referees. The changes regarding points 1) and 3) of Reviewer # 2 and my own point number 1 seem completely satisfactory. With regard to point 2) of Reviewer # 2, which is intimately related to my points 2 and 3, the authors admitted its relevance, but decided against its detailed discussion. However, since all three referees agree that this is a fundamental result, I recommend publishing the paper in its present version. Uli Buchenau

We thank the referee for the positive review. We have not made any changes following this comment, but plan to enter the points addressed by the referees in an upcoming more specialized paper.